# General OCR Theory: Towards OCR-2.0 via a Unified End-to-end Model

## Abstract

Traditional OCR systems (OCR-1.0) are increasingly unable to meet people's usage due to the growing demand for intelligent processing of man-made optical characters. In this paper, we collectively refer to all artificial optical signals (e.g., plain texts, math/molecular formulas, tables, charts, sheet music, and even geometric shapes) as "characters" and propose the **G**eneral **O**CR **T**heory along with an excellent model, namely GOT, to promote the arrival of OCR-2.0. The GOT, with 580M parameters, is a unified, elegant, and end-to-end model, consisting of a high-compression encoder and a long-contexts decoder. As an OCR-2.0 model, GOT can handle all the above "characters" under various OCR tasks. On the input side, the model supports commonly used scene- and document-style images in slice and whole-page styles. On the output side, GOT can generate plain or formatted results (markdown/tikz/smiles/kern) via an easy prompt. Besides, the model enjoys interactive OCR features, i.e., region-level recognition guided by coordinates or colors. Furthermore, we also adapt dynamic resolution and multi-page OCR technologies to GOT for better practicality. In experiments, we provide sufficient results to prove the superiority of our model.

## 1 Introduction

Optical Character Recognition (OCR) is a widely used technology that extracts the characters embedded in an optical image into an editable format. Typical OCR systems Du et al. (2021) in the OCR-1.0 era are mainly designed based on a multi-modular pipeline style, commonly including element detection, region cropping, and character recognition parts. Each module is prone to falling into local optima, making the whole system incur high maintenance costs. Moreover, traditional OCR methods have insufficient general ability, reflected as different OCR-1.0 networks usually designed for different sub-tasks. Nevertheless, choosing a suitable one from diverse OCR models for a special task is always inconvenient for users.

In the past year, Large Vision Language models (LVLMs) OpenAI (2023); Liu et al. (2023b); Ye et al. (2023a) have developed rapidly and showcased impressive performance. As a highly anticipated ability, the OCR performance of current LVLMs is continuously improving. Based on CLIP Radford et al. (2021), LLaVA Liu et al. (2023b) naturally acquires the English OCR ability after the instruct tuning phase. To lift the OCR accuracy and support other languages, e.g., Chinese, Qwen-VL Bai et al. (2023b) unfreezes its image encoder (a CLIP-G) and uses lots of OCR data in its stage-two training. Innovatively, Vary Wei et al. (2023) generates a new high-resolution OCR vision vocabulary paralleling the CLIP branch to deal with document-level dense OCR. By contrast, InternVL-1.5 Chen et al. (2024b) and other models Liu et al. (2024d); Ye et al. (2023b) utilize a sliding window manner to crop the whole image into multiple sub-patches for high-resolution OCR. Hence, a consensus is that optical character perception and recognition are the foundation of text-driven image understanding, drawing many researchers to pay more attention to LVLMs' OCR booster.

However, the popular designs of LVLMs may not be suitable for diverse OCR tasks for the following reasons: 1) The conflicts between perception and reasoning. LVLMs mainly focus on visual reasoning performance, e.g., VQA Singh et al. (2019); Mathew et al. (2021), because that is what the LLM excels at. To quickly obtain the QA-gain benefits from LLMs, most LVLMs Liu et al. (2023b); Ye et al. (2023a); Li et al. (2023a) align image tokens to text ones. However, it is unreasonable to do this for pure perception OCR tasks, especially high-density text scenes, because each aligned vision

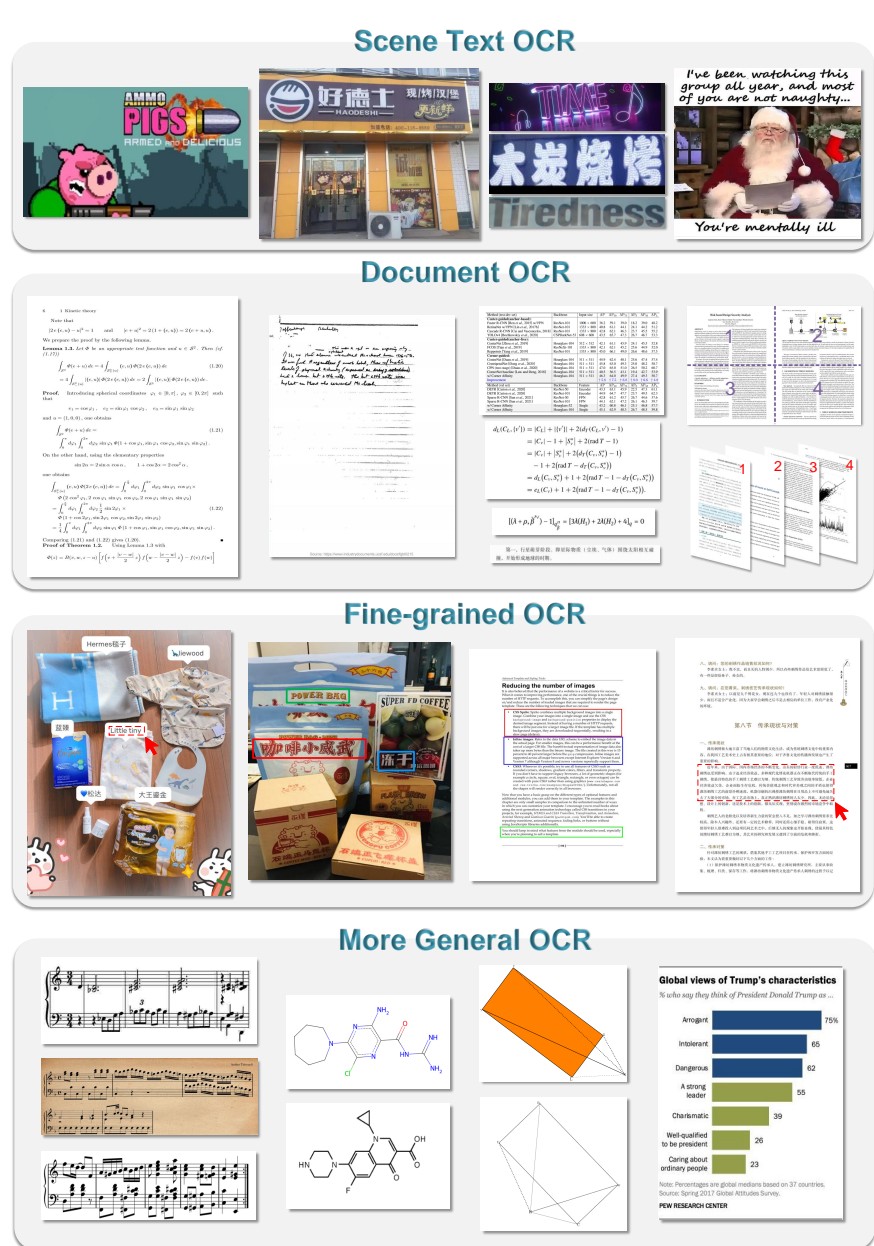

Figure 1: On the input side, GOT supports various optical image types, such as commonly used photographs and documents. Besides, as a general OCR-2.0 model, GOT can handle more tasks, e.g., sheet music, molecular formulas, easy geometric shapes, charts, etc. Moreover, the model can adapt to region-focus OCR, high-resolution OCR, and multiple-page OCR. GOT mainly supports English and Chinese and can control the structure results (Mathpix markdown/tikz/smiles/kern) via a prompt.

token (biased towards text token) cannot compress enough characters. Imagine how wasteful it is to use thousands of image tokens, e.g., the image-cropping manner Chen et al. (2024b); Liu et al. (2024c), to encode an equal amount of optical characters (e.g., texts within only an A4-PDF page). 2) High iteration and deployment costs. LVLM often enjoys billions of parameters, leading to the post-training and deployment costs being too high. Generally speaking, for LVLMs, fine-tuning is not enough once we want to add a new OCR pattern, e.g., a new language, instead of enough GPU resources for pre-training. However, rerunning the pre-training with billions of parameters, only to introduce a new OCR feature, is also wasteful.

Accordingly, we propose the general OCR theory, i.e., OCR-2.0, to break the bottlenecks of both traditional and LVLM manners on OCR tasks. We think that a model of OCR 2.0 should have the following essential characteristics:

- **End-to-end.** Compared to OCR-1.0 models with complex procedures, the OCR-2.0 model should enjoy a unified and end-to-end architecture to ensure lower maintenance costs. It is cool that a beginner can quickly master the entire OCR system in the 2.0 era.

- **Low training and inference costs.** The OCR-2.0 model should not be a chatbot, like LVLM, that focuses on reasoning tasks. Its focus should be on strong perception and recognition of optical characters, so it needs a reasonable number of model parameters in exchange for lower training and inference costs.

- **Versatility.** The OCR-2.0 model's other important point is versatility, including recognizing more general artificial optical "characters", e.g., sheet music, charts, geometric shapes, etc. Besides, the model should support the output format with stronger readability, e.g., LaTeX/Markdown format for formulas and tables.

Based on the proposed general OCR theory, we present a primary OCR-2.0 model (GOT) to bridge the gap between OCR-1.0 models and people's higher optical character processing demands. In architecture, we adopt the unsophisticated encoder-decoder paradigm for the model. Specifically, GOT enjoys a high compression rate encoder to transfer the optical image to tokens as well as a long context length decoder to output the corresponding OCR results. The encoder has approximately 80M parameters posing $1024\times1024$ input size which is enough to deal with commonly used photo/document input styles. Each input image will be compressed to tokens with $256\times1024$ dimensions. The decoder of GOT, with 0.5B parameters, supports 8K max length tokens to ensure it can tackle long-context scenarios. We devise an effective and efficient training strategy for GOT, which can be divided into three procedures, i.e., decoupled pre-training of the encoder, joint-training of the encoder with a new decoder, and further post-training of the decoder. Besides, to further lift the practicality of GOT, we additionally adapt the fine-grained OCR feature for better interactivity, dynamic resolution strategy for ultra-high-resolution images (e.g., over 2K), and the multi-page OCR technology to alleviate the problem of difficulty in breaking pages in PDF image-text pairs (e.g., page breaks in *.tex* files). To support each training stage, we do many data engines for synthetic data production, which is the key to the success of GOT and will be described in detail in this paper. The main input data format supported by our model can be seen in Figure 1.

As a model for envisioning OCR-2.0, GOT demonstrates promising performance in our experiments in various OCR tasks. We hope the proposed simple and elegant GOT can draw more researchers to invest in the research of OCR-2.0. Of course, the path to OCR-2.0 is still long and GOT also enjoys much improvement room, such as supporting more languages, more general artificial signals, and more complex geometries. In this new era led by LVLMs, we are convinced that the pure OCR model is not over, it may even be a new beginning.

## 2 RELATED WORK

### 2.1 TRADITIONAL OCR

Optical Character Recognition (OCR) is a classic research topic that aims to convert the image's optical contents into an editable format for further downstream processing. Traditional OCR systems, called OCR-1.0, typically use a framework that is assembled from multiple expert modules. For instance, to handle diverse optical characters, the OCR system Du et al. (2021) is usually developed by integrating several domain expert networks, such as layout analysis Zhong et al. (2019), text detection Liao et al. (2022); Liu et al. (2019b); Zhang et al. (2021), region extraction, and contents recognition Li et al. (2023b). The reason for using such a pipeline scheme is that the text recognition module (the OCR part) failed to scale up successfully, which can only deal with the image format of small slices, resulting in the entire OCR process being in the form of first detecting texts/cropping regions, and then recognizing the results within the slice. However, a system with complicated procedures may suffer potential systematic errors and high maintenance costs. Although some OCR-1.0 models, e.g., Nougat Blecher et al. (2023) can directly process documents at the whole page level, they are often designed and trained for a specific sub-task, leading to unsatisfactory general

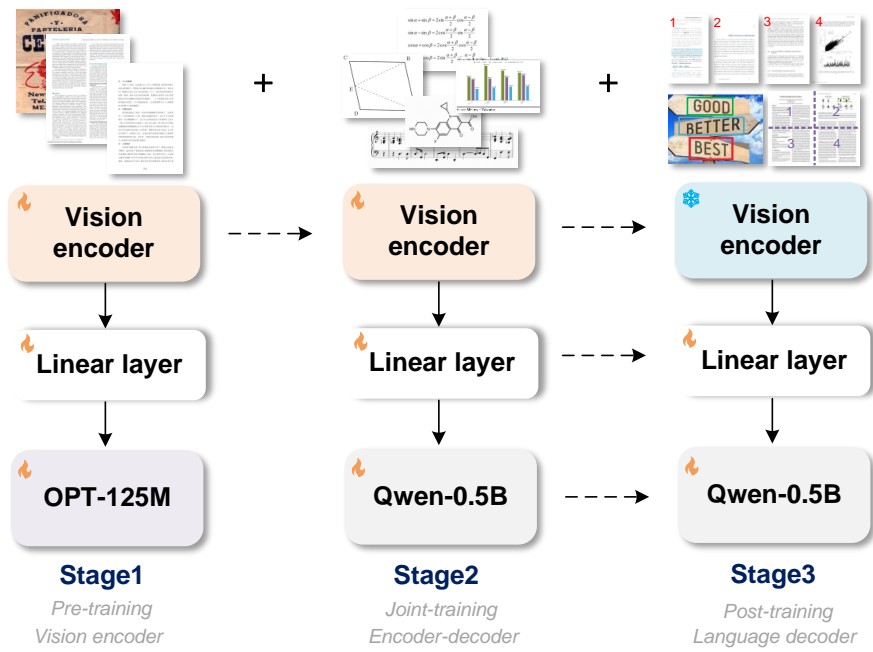

Figure 2: The framework of the proposed GOT. Stage 1: We pre-train the vision encoder using a tiny OPT-125M to adapt the OCR tasks efficiently. Stage 2: GOT is built by connecting the vision encoder to Qwen-0.5B and sufficient OCR-2.0 knowledge of more general optical characters is used in this stage. Stage 3: No modification of the vision encoder is required, and GOT is customized to new character recognition features.

ability. In the OCR-1.0 era, one inconvenient thing is that we usually need to switch different models according to various OCR needs.

## 2.2 LVLM-DRIVEN OCR

Large Vision-Language Models Liu et al. (2023b); Bai et al. (2023b); Wei et al. (2023); Ye et al. (2023a); Chen et al. (2024b); Liu et al. (2024d;a) have attracted lots of attention in the AI-community due to their powerful generalization capabilities. For the current LVLMs owning perception-reasoning comprehensive capacity, the OCR ability has become a hot spot with the increasing demand for text-driven visual understanding. Most LVLMs' OCR capabilities come from the ready-made CLIP Radford et al. (2021), especially those that freeze CLIP encoder Liu et al. (2023b) to complete the entire LVLM training. For such models, the vanilla CLIP, mainly with English scene text knowledge, is the bottleneck for the OCR performance to out-of-domain tasks, such as other languages or documents. Some other LVLMs Ye et al. (2023a); Bai et al. (2023b) choose to unfreeze the encoder and freeze the LLM for training to enhance the CLIP-encoder and align the image tokens to text ones. These models will face the problem of low optical character compression rate, as it is difficult for frozen LLM to decode too much text from an aligned image token. To alleviate this problem, some models Chen et al. (2024b); Liu et al. (2024d); Ye et al. (2023b) adopt a sliding window manner to decompose input images into smaller patches. Although this dynamic resolution approach is highly effective in processing high-resolution input images, e.g., PDF, it will result in excessive image tokens and limit the max length of the generated OCR result to some extent.

## 3 GENERAL OCR THEORY

In this work, we propose the general OCR theory, i.e., OCR-2.0 (as expounded in Section 1) to promote the development of the OCR field. Based on the proposed new theory, we present a novel OCR model (GOT). In this section, we will introduce the technical details of our model.

## 3.1 Framework

As illustrated in Figure 2, GOT comprises three modules, i.e., an image encoder, a linear layer, and an output decoder. The linear layer acts as the connector to map the channel dimension between the vision encoder and the language decoder. We utilize three main steps in optimizing the whole GOT model. First, we conduct the pure text recognition task to pre-train the vision encoder. To lift training efficiency and save GPU resources, we choose a tiny decoder to pass gradients to the encoder. In this stage, we feed images containing scene texts and manual images containing document-level characters into the model to allow the encoder to gather the two most commonly used characters' encoding abilities. In the next stage, we form the architecture of GOT by connecting the trained vision encoder to a new larger decoder. We prepare lots of more general OCR data (*e.g.*, sheet music, math/molecular formulas, and geometric shapes) to scale up the OCR-2.0 knowledge for this stage. In the final stage, we intend to improve the generalization and applicability of GOT further. Specifically, fine-grained and muti-crop/page synthetic data are generated and added for GOT to support region prompt OCR Liu et al. (2024a), huge image OCR, and batched PDF OCR features.

## 3.2 Pre-train the OCR-earmarked Vision Encoder

As aforementioned, GOT enjoys the encoder-decoder structure. Inspired by the LVLMs design, the decoder can be initialized by a well-trained language model. However, we did not find a suitable pre-trained encoder for an OCR-2.0 model, so we must train one ourselves. We hope the new OCR encoder can work well on commonly used scene and document text recognition in various input shapes (both slices and whole pages).

### 3.2.1 The Vision Encoder Generation.

The encoder structure we selected is VitDet Li et al. (2022) (base version with about 80M parameters) due to its local attention can greatly reduce the computational cost of high-resolution images. We follow the Vary-tiny setting Wei et al. (2023) to design the last two layers of the encoder, which will transfer a $1024 \times 1024 \times 3$ input image to $256 \times 1024$ image tokens. Then, these image tokens are projected into language model (OPT-125M Zhang et al. (2022)) dimension via a $1024 \times 768$ linear layer. Unlike the Vary encoder which only focuses on a single document task under a relatively unitary input shape, we incorporated natural scenes and cropped slices during our pre-training. In the pre-processing stage, images of each shape are directly resized to $1024 \times 1024$ squares, as square shapes can be used to adapt to images of various aspect ratios with a compromise.

### 3.2.2 Data Engine Towards Encoder Pre-training

In such an encoder pre-training stage, we use about 5M image-text pairs, including 3M scene text OCR data and 2M document OCR data. Their acquisition methods are as follows:

For the natural scene data, the English/Chinese images are sampled from Laion-2B Schuhmann et al. (2022) and Wukong Gu et al. (2022) datasets, respectively. Then, the pseudo ground truth in these diverse real scenes is captured using PaddleOCR Du et al. (2021) tools. Overall, we obtain 2M dat with half in Chinese and half in English. For text ground truth, we perform two types of processing: 1) remove the bounding box and combine each text content in order from top to bottom and left to right. 2) crop the text region from the original image according to the bounding box and save it as image slices. The later method 2) allows us to obtain another 1M slice-type image-text pairs.

For the document-level data, we first collect open-source PDF-style files from the Common Crawl and employ the Fitz Python package to extract corresponding dense text content. In such a process, we gain 1.2M full-page PDF-style image-text pairs and 0.8M image slice data. The slice data, including line- and paragraph-level, is cropped from the PDF image via the parsed bounding box.

## 3.3 Scaling Up the OCR-2.0 Knowledge via Multi-task Joint-training

### 3.3.1 The Final Architecture of GOT

After the pre-training step of the vision encoder, we connect it to a larger language model with more powerful capabilities to build the final architecture of GOT. Here, we adopt the Qwen Bai et al.

(2023a) with 500M parameters as the decoder because it has a relatively small number of parameters while incorporating prior knowledge of multiple languages. The dimension of the connector (i.e., the linear embedding layer) is adjusted into 1024×1024 to align with the input channels of the Qwen-0.5B. Hence, GOT enjoys the seamless encoder-decoder paradigm with about 580M parameters in total, which is more computationally resource-friendly and easier to deploy on a consumer-grade GPU with 4G memory. The high compression rate (1024×1024 optical pixels to 256 image tokens) of the encoder saves a lot of token space for the decoder to generate new tokens. Meanwhile, the satisfactory decoding context length (we use about 8K max-length) of the decoder ensures that the GOT can effectively output OCR results under dense scenes.

### 3.3.2 DATA ENGINE FOR JOINT-TRAINING

To inject sufficient OCR-2.0 knowledge into GOT, instead of the above-mentioned plain OCR data, we carefully explore several synthesis methods and data engines in this stage, as shown in Figure 3. We will delve into the details of each type of synthetic data in the following paragraphs.

**Plain OCR data.** We use 80% of the data mentioned in Section 3.2.2 as plain OCR data. To further enhance the robustness of GOT, we also add the handwritten text recognition sub-task, which involves various styles of handwriting from letters and diaries in different languages. We collect the Chinese CASIA-HWDB2 Tek (2024a), English IAM Tek (2024b), and Norwegian NorHand-v3 Tek (2024c) datasets to meet our requirements. For the original image-text pairs with the line-level slice format, 6∼8 pairs are grouped and randomly pasted into a blank document page to achieve longer-text handwriting recognition and improve training efficiency.

**Mathpix-markdown formatted data.** Preserving the optical content format is critical to maintaining strong readability for the output results, especially for mathematical formulas and tables. To this end, we use multiple approaches to gather as much formatted data as possible. The details of data collection and production are as follows:

- **Math formulas.** We crawl a large number of LATEX source *.tex* files on Arxiv and extract about 1M formula fragments from them. Next, we transfer the formula sources to Mathpix format and use the Chorme-driver to call Mathpix-markdown-it tool to render the sources to HTML format. We then convert the HTML files to SVGs and save them as PNG images. We find that this rendering method is more than 20× faster than directly using the LATEX.

- **Molecular formulas.** We first download the *ChEMBL_25* file that contains 2M smile sources. Then we use the Mathpix-markdown-it tool and *rdkit.Chem* package to gather about 1M of molecular formula image-text pairs.

- **Table**. From the crawled *.tex* files, we extract about 0.3M table sources and render them into images. Instead of Mathpix-markdown-it, we directly utilize the LATEX as the rendering tool due to its better rendering effects for advanced tables.

- **Full page data.** Using the Nougat Blecher et al. (2023) method, we obtain about 0.5M English markdown PDF-text pairs. Besides, following Vary Wei et al. (2023; 2024), we gather another 0.5M Chinese markdown pairs. We transfer their contents to Mathpix format. Furthermore, we additionally add 0.2M in-house data, which is directly labeled using Mathpix, including books, papers, and financial reports.

**More general OCR data.** We hope GOT can deal with more general optical artificial "characters". Accordingly, we collect three related challenging tasks and generate the corresponding data. They are sheet music, geometric shapes, and charts, respectively.

- **Sheet music.** Music is a precious part of the cultural heritage and optical music recognition Calvo-Zaragoza et al. (2020); Ríos-Vila et al. (2024) plays an important role in achieving automatic recognition and transcription of sheet music. We choose the GrandStaff Ríos-Vila et al. (2023) dataset as the source to render. The dataset of polyphonic music scores provides the *Humdrum* **\*\*kern** transcriptions from the excerpts of music. In addition to the existing approximately 100K image-text samples, we also extract some text samples to re-render via the Verovio Python Package. We mainly add new backgrounds from white to real paper styles and randomly add the title and

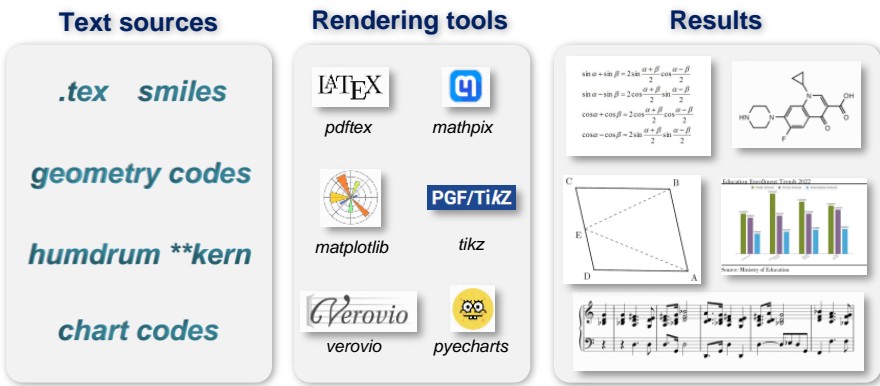

Figure 3: We use six rendering tools to run data engines to make the GOT work well on diverse OCR tasks. We utilize the LATEX for tables, Mathpix-markdown-it for math/molecular formulas, Tikz for simple geometric shapes, Verovio for sheet music, and Matplotlib/Pyecharts for charts, respectively.

author information. Note that we only render single-system sheet music due to we don't have professionals in the relevant field and we do not know how to assemble single-system sheets to a full page. After rendering, we collect about 0.5M samples.

- **Geometric shape.** Geometry is a key capability of LVLMs and is a necessary step towards AGI. GOT is expected to transform optical geometric elements into TikZ Mertz & Slough (2007) text format. TikZ contains some concise commands to produce basic geometric elements and they can be compiled using LATEX. We employ TikZ-style points and lines and use the simplest point-line spatial relationship to construct simple basic geometric shapes (*e.g.*, circles, rectangles, triangles, and combined shapes) as well as simple function curves (*e.g.*, straight lines, parabolas, ellipses, hyperbolas, and so on). Through this method, we obtained approximately 1M geometric Tikz data. The geometric rendering is complicated, and our current work is only a preliminary attempt. GOT can only recognize basic geometry at present, yet we believe that with the development of synthetic data technology and OCR-2.0, future models will be able to identify complex geometric shapes.

- **Chart.** Charts are crucial in data visualization and data analysis of several research fields. The proposed GOT refers to the chart structural extraction sub-task as "Chart OCR", which converts the visual knowledge (*e.g.*, title, source, x-title, y-title, and values) on the chart image into an editable output with a table/Python-dict format. Following OneChart Chen et al. (2024a), the chart image-text pairs are rendered using Matplotlib and Pyecharts tools. Because GOT is only an OCR model, we don't need the elements of the chart synthesized to be semantically related. Thus, we just randomly extract entity texts (for the title, source, x-title, y-title, etc) from the open-access NLP corpus. The numerical values are random numbers under a controlled distribution. Through this method, we obtained 2M chart data, with half from Matplotlib and half from Pyecharts.

### 3.4 CUSTOMIZING NEW OCR FEATURES BY POST-TRAINING THE DECODER

After compressing the general visual information of the diverse OCR-2.0 optical signals via the above two steps, GOT is ready to perform image-level OCR tasks in various scenarios. Based on this perceptually savvy vision encoder, GOT can be easily tuned to meet the users' needs for input and output. Here, we customize GOT to enable three new features, i.e., fine-grained, multi-page, and dynamic resolution OCR, by only post-training the decoder part.

#### 3.4.1 FINE-GRAINED DATA ENGINE FOR INTERACTIVE OCR

As a high-interactivity feature, fine-grained OCR Liu et al. (2024a) is the region-level visual perception controlled by spatial coordinates or colors. The user can add box coordinates or color text in the question prompt to request recognition within the region of interest (RoI), avoiding the output of other irrelevant characters. For the natural fine-grained OCR, the source images are from opensource datasets, including RCTW Shi et al. (2017), ReCTS Liu et al. (2019a), and ShopSign Zhang et al. (2019), and COCO-Text Veit et al. (2016) dataset. The datasets mentioned above provide the text

bounding boxes, so we can use them to produce fine-grained (region/color prompt) OCR data directly. For the document-level fine-grained OCR, following Fox Liu et al. (2024a), we filter out those with the scanned format in the downloaded PDF files and parse the left part using Python packages (Fitz/PDFminer). We record the page-level images, bounding boxes of each line/paragraph, and the corresponding texts to produce the ground truth of the box-guided OCR sub-task. For such a task, each coordinate value is first normalized and then magnified 1000 times. For the color-guided task, we choose the most commonly used colors (red, green, and blue) as the frame colors and draw them via the corresponding bounding box on the original image. Overall, we gather about 600K samples.

### 3.4.2 Multi-crop Data Engine for Ultra-large-image OCR

GOT supports 1024×1024 input size, which is enough for commonly used OCR tasks, e.g., scene OCR or A4-page PDF OCR. However, dynamic resolution is required for some scenes with huge images, such as two-page PDF horizontal stitching (commonly occurring when reading papers). Thanks to our high compression rate encoder, the dynamic resolution of GOT is achieved under a large sliding window (1024×1024), ensuring that our model can complete extreme resolution OCR tasks with acceptable image tokens. We use the InternVL-1.5 Chen et al. (2024b) cropping method with tiles max to 12. The ultra-resolution images are synthesized using the single-page PDF data mentioned above, including horizontal and vertical stitching, leading to 500K image-text pairs.

### 3.4.3 Multi-page Data Engine for Batched PDF-file OCR

For OCR tasks, it is reasonable to use a "for loop" for multi-page processing. We introduce the multi-page OCR (without "for loop") feature for GOT due to some formatted PDF data making it difficult to break pages (to obtain text that is completely incompatible with each page) to further scale up, such as *.tex* in Arxiv. We hope that with GOT, researchers no longer have to worry about PDF ground truth page breaks (e.g., Nougat Blecher et al. (2023)), as they can train on multiple pages directly. To realize such a feature, we randomly sample 2-8 pages from our Mathpix formatted PDF data and join them together to form a single round OCR task. Each selected page contains text that is less than 650 tokens, to ensure that the overall length does not exceed 8K. In total, we generate about 200K multi-page OCR data, most of which are interlaced between Chinese and English pages.

## 4 Experiments

### 4.1 Implement Details

We use 8×8 L40s GPUs to train GOT. In the pre-training stage, we optimize all parameters with a global batch size of 128 and train for 3 epochs. We utilize the AdamW Loshchilov & Hutter (2019) optimizer and a cosine annealing scheduler Loshchilov & Hutter (2016) with a start learning rate of 1e-4. The max token length in this stage is set to 4096. In the joint-training stage, we put the max token length to 6000 and train the model with the same optimizer settings as stage 1 for 1 epoch. In the last post-training stage, we expand the max token length to 8192 to allow the model to support multi-patch/page OCR features. In this stage, the beginning learning rate is 2e-5, and the epoch is 1. During each train-data process, 80% of the data from the previous stage is sampled for the following stage to ensure that the basic ability does not degrade when adding new features.

### 4.2 Main Results

In this section, we verify the performance of GOT on 5 different OCR tasks, including 1) plain document OCR; 2) scene text OCR; 3) fine-grained document OCR; 4) formatted (Mathpix markdown) document OCR; 5) more general character OCR. Note that the test data for each benchmark undergoes strict text similarity filtering to ensure that it is not included in the training data. Sources of each test benchmark and model performance analysis are as follows.

### 4.2.1 Plain document OCR performance

We use the open-source Fox Liu et al. (2024a) benchmark to test the performance of GOT and popular LVLMs on both Chinese and English PDF OCR. The metrics we used are those commonly in OCR

Table 1: Performance comparison of dense English (en) and Chinese (zh) OCR on document-level pages. The results of other models are from the previous work Liu et al. (2024a).

| Method | Size | Edit Distance↓ | | F1-score↑ | | Precision↑ | | Recall↑ | | BLEU↑ | | METEOR↑ | |
|---|---|---|---|---|---|---|---|---|---|---|---|---|---|
| | | en | zh | en | zh | en | zh | en | zh | en | zh | en | zh |
| UReader Ye et al. (2023b) | 7B | 0.718 | - | 0.344 | - | 0.296 | - | 0.469 | - | 0.103 | - | 0.287 | - |
| LLaVA-NeXT Liu et al. (2024c) | 34B | 0.430 | - | 0.647 | - | 0.573 | - | 0.881 | - | 0.478 | - | 0.582 | - |
| InternVL-ChatV1.5Chen et al. (2024b) | 26B | 0.393 | 0.265 | 0.751 | 0.816 | 0.698 | 0.784 | 0.917 | 0.866 | 0.568 | 0.622 | 0.663 | 0.717 |
| Nougat Blecher et al. (2023) | 250M | 0.255 | - | 0.745 | - | 0.720 | - | 0.809 | - | 0.665 | - | 0.761 | - |
| TextMonkey Liu et al. (2024d) | 7B | 0.265 | - | 0.821 | - | 0.778 | - | 0.906 | - | 0.671 | - | 0.762 | - |
| DocOwl1.5 Hu et al. (2024) | 7B | 0.258 | - | 0.862 | - | 0.835 | - | 0.962 | - | 0.788 | - | 0.858 | - |
| Vary Wei et al. (2023) | 7B | 0.092 | 0.113 | 0.918 | 0.952 | 0.906 | 0.961 | 0.956 | 0.944 | 0.885 | 0.754 | 0.926 | 0.873 |
| Vary-toy Wei et al. (2024) | 1.8B | 0.082 | 0.142 | 0.924 | 0.914 | 0.919 | 0.928 | 0.938 | 0.907 | 0.889 | 0.718 | 0.929 | 0.832 |
| Qwen-VL-Plus Bai et al. (2023b) | - | 0.096 | 0.121 | 0.931 | 0.895 | 0.921 | 0.903 | 0.950 | 0.890 | 0.893 | 0.684 | 0.936 | 0.828 |
| Qwen-VL-Max Bai et al. (2023b) | >72B | 0.057 | 0.091 | 0.964 | 0.931 | 0.955 | 0.917 | **0.977** | 0.946 | 0.942 | 0.756 | **0.971** | 0.885 |
| Fox Liu et al. (2024a) | 1.8B | 0.046 | 0.061 | 0.952 | 0.954 | 0.957 | 0.964 | 0.948 | 0.946 | 0.930 | 0.842 | 0.954 | 0.908 |
| **GOT** | 580M | **0.035** | **0.038** | **0.972** | **0.980** | **0.971** | **0.982** | 0.973 | **0.978** | **0.947** | **0.878** | 0.958 | **0.939** |

tasks, i.e., edict distance, F1-score, precision, recall, BLEU, and METEOR. Due to the lengthy text of the document, we use word-level segmentation to calculate each indicator. As shown in Table 1, with only 580M, GOT achieves advanced performance on pure text OCR in the document, proving the excellent PDF text perception and recognition ability.

Table 2: Performance of English (en) and Chinese (zh) OCR for scene texts. On these common image-level OCR tasks, GOT can achieve better results compared to other popular models.

| Method | Size | Edit Distance↓ | | F1-score↑ | | Precision↑ | | Recall↑ | | BLEU↑ | | METEOR↑ | |
|---|---|---|---|---|---|---|---|---|---|---|---|---|---|
| | | en | zh | en | zh | en | zh | en | zh | en | zh | en | zh |
| UReader Ye et al. (2023b) | 7B | 0.568 | - | 0.661 | - | 0.843 | - | 0.569 | - | 0.258 | - | 0.488 | - |
| LLaVA-NeXT Liu et al. (2024c) | 34B | 0.499 | - | 0.558 | - | 0.637 | - | 0.538 | - | 0.379 | - | 0.678 | - |
| TextMonkey Liu et al. (2024d) | 7B | 0.331 | - | 0.743 | - | 0.827 | - | 0.710 | - | 0.521 | - | 0.728 | - |
| DocOwl1.5 Hu et al. (2024) | 7B | 0.334 | - | 0.788 | - | 0.887 | - | 0.751 | - | 0.525 | - | 0.708 | - |
| InternVL-ChatV1.5Chen et al. (2024b) | 26B | 0.267 | 0.123 | 0.834 | 0.913 | **0.942** | **0.934** | 0.790 | 0.902 | 0.587 | 0.588 | 0.744 | 0.876 |
| Qwen-VL-Max Bai et al. (2023b) | >72B | 0.182 | 0.168 | 0.881 | 0.867 | 0.891 | 0.878 | 0.888 | 0.873 | 0.586 | 0.572 | 0.848 | 0.845 |
| **GOT** | 580M | **0.112** | **0.096** | **0.926** | **0.928** | 0.934 | 0.914 | **0.927** | **0.954** | **0.676** | **0.641** | **0.896** | **0.928** |

### 4.2.2 SCENE TEXT OCR PERFORMANCE

We collect 400 natural images, half in Chinese and half in English, as the scene text OCR benchmark. All the ground truth in this benchmark are manually corrected. Because the text in the scene image is relatively short, we use character-level segmentation to calculate various metrics. As shown in Table 2, we can see that GOT also works well on natural images, demonstrating the model's excellent performance on most basic OCR tasks (both document and scene texts).

### 4.2.3 FORMATTED DOCUMENT OCR PERFORMANCE

Converting the optical PDF image to a markdown-like format is an important feature of an OCR model. To verify this ability of GOT, we carefully prepare 90 pages of samples as a high-quality benchmark. The benchmark, containing both Chinese and English document pages, is first generating pseudo-labels via Mathpix, and then manually correcting for errors. In Table 3, we can see the single-scale (1024×1024) GOT can yield satisfactory results. When we use multi-crop inference, the performance of GOT is further lifted especially on formulas and tables with small texts. The results prove the effectiveness of GOT on documents with formatted outputs. Besides, the dynamic resolution scheme is a good choice when processing higher-resolution images.

### 4.2.4 FINE-GRAINED OCR PERFORMANCE

We report the fine-grained OCR metrics of GOT. As shown in Table 4, the GOT is overall better than Fox Liu et al. (2024a) on both the bounding box-based and color-based referential OCR tasks, indicating that our model enjoys excellent interactive OCR capabilities.

Table 3: Performances of formatted document (Chinese/English) and more general OCR. Single means the input is the vanilla image and multi-crop represents the dynamic resolution strategy.

|  | Types | Edit Distance↓ | F1-score↑ | Precision↑ | Recall↑ | BLEU↑ | METEOR↑ |
|---|---|---|---|---|---|---|---|
| Markdown document | **single:** | | | | | | |
|  | All text | 0.097 | 0.942 | 0.944 | 0.942 | 0.877 | 0.876 |
|  | Formula | 0.269 | 0.749 | 0.771 | 0.751 | 0.512 | 0.716 |
|  | Table | 0.254 | 0.867 | 0.857 | 0.897 | 0.756 | 0.760 |
|  | **muti-crop:** | | | | | | |
|  | All text | 0.086 | 0.953 | 0.948 | 0.960 | 0.896 | 0.903 |
|  | Formula | 0.159 | 0.865 | 0.858 | 0.882 | 0.628 | 0.828 |
|  | Table | 0.220 | 0.878 | 0.861 | 0.919 | 0.779 | 0.811 |
| Geneal | Sheet music | 0.046 | 0.939 | 0.963 | 0.939 | 0.900 | 0.923 |
|  | Geometry | 0.061 | 0.884 | 0.882 | 0.888 | 0.766 | 0.882 |

Table 4: Comparison of fine-grained document OCR. Without the need to tune the vision encoder, GOT can easily achieve excellent capabilities of box-guided OCR and color-guided OCR.

| **Metrics** | **English** | | | | | **Chinese** | | | |
|---|---|---|---|---|---|---|---|---|---|
|  | box | | | color | | box | | color | |
|  | DocOwl1.5 | Fox | GOT | Fox | GOT | Fox | GOT | Fox | GOT |
| Edit Distance ↓ | 0.435 | 0.059 | **0.041** | 0.064 | **0.034** | 0.042 | **0.033** | 0.114 | **0.040** |
| F1-score ↑ | 0.670 | 0.957 | **0.970** | 0.940 | **0.966** | 0.955 | **0.965** | 0.884 | **0.957** |
| Precision ↑ | 0.886 | 0.962 | **0.973** | 0.942 | **0.970** | 0.966 | **0.974** | 0.902 | **0.969** |
| Recall ↑ | 0.617 | 0.955 | **0.969** | 0.942 | **0.964** | 0.947 | **0.958** | 0.873 | **0.948** |
| BLEU ↑ | 0.478 | 0.914 | **0.926** | 0.868 | **0.910** | 0.885 | **0.898** | 0.778 | **0.884** |
| METEOR ↑ | 0.569 | 0.955 | **0.966** | 0.938 | **0.961** | 0.934 | **0.942** | 0.848 | **0.931** |

Table 5: Performance on number-centric chart OCR. With sufficient optimization of visual perception and dense information compression, GOT surpasses the popular models by a large margin.

|  | Metric | Deplot (1.3B) | UniChart (0.26B) | ChartVLM (7.3B) | GPT-4V (>100B) | Qwen-VL (>72B) | **GOT** (0.58B) |
|---|---|---|---|---|---|---|---|
| ChartQA-SE | AP@strict | 0.614 | 0.423 | 0.718 | 0.504 | 0.586 | **0.747** |
|  | AP@slight | 0.709 | 53.18 | 0.814 | 0.606 | 0.685 | **0.845** |
|  | AP@high | 0.729 | 0.560 | 0.842 | 0.643 | 0.727 | **0.867** |
| PlotQA-SE | AP@strict | 0.031 | 0.105 | 0.038 | 0.073 | 0.005 | **0.133** |
|  | AP@slight | 0.165 | 0.260 | 0.468 | 0.194 | 0.042 | **0.596** |
|  | AP@high | 0.265 | 0.269 | 0.540 | 0.223 | 0.120 | **0.640** |

### 4.2.5 MORE GENERAL OCR PERFORMANCE

We utilize the sheet music, geometry, and chart benchmarks to verify GOT's more general OCR performance. For the first two tasks, we separately render 100 and 180 additional samples as benchmarks, and as can be seen in Table 3, GOT still performs well on these new OCR tasks. For chart OCR, we use structure-extraction version Chen et al. (2024a) ChartQA Masry et al. (2022) and PlotQA Methani et al. (2020) as benchmarks. In Table 5, the chart OCR ability of GOT is even much better than the chart-specific models Liu et al. (2023a); Masry et al. (2023); Xia et al. (2024) and popular LVLMs OpenAI (2023); Bai et al. (2023b). All results demonstrate the effectiveness of our model on more general OCR tasks.

## 5 CONCLUSION

This paper presents a primary OCR-2.0 model that is structurally simpler than OCR-1.0 systems, focuses more on pure OCR tasks than LVLMs, and enjoys superior performance. OCR-2.0 integrates various pan-OCR tasks into one model and is a valuable research direction in model design, data engineering, and application scenarios. We want the simple, elegant, effective, and promising GOT OCR-2.0 model to attract more attention to such a task.

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

# A  APPENDIX

In this section, we provide sufficient output results of GOT to show its outstanding OCR performance. We also demonstrate the format of the corresponding input prompt for different types of OCR tasks.

**Prompt: OCR with format:**

6        1 Kinetic theory

Note that

$$|2e(e,u) - u|^2 = 1 \quad \text{and} \quad |e+u|^2 = 2(1+(e,u)) = 2(e+u,u).$$

We prepare the proof by the following lemma.

**Lemma 1.3.** Let $\Phi$ be an appropriate test function and $u \in \mathcal{S}^2$. Then (cf. (1.17))

$$\int_{\mathcal{S}^2} \Phi(e+u)\,de = 4\int_{\mathcal{S}^2_+(u)} (e,u)\Phi(2e(e,u))\,de \qquad (1.20)$$

$$= 4\int_{\mathcal{S}^2_+(u)} |(e,u)|\Phi(2e(e,u))\,de = 2\int_{\mathcal{S}^2} |(e,u)|\Phi(2e(e,u))\,de.$$

**Proof.**  Introducing spherical coordinates $\varphi_1 \in [0,\pi]$, $\varphi_2 \in [0,2\pi]$ such that

$$e_1 = \cos\varphi_1, \quad e_2 = \sin\varphi_1\cos\varphi_2, \quad e_3 = \sin\varphi_1\sin\varphi_2$$

and $u = (1,0,0)$, one obtains

$$\int_{\mathcal{S}^2} \Phi(e+u)\,de = \qquad (1.21)$$

$$\int_0^\pi d\varphi_1 \int_0^{2\pi} d\varphi_2 \sin\varphi_1\Phi(1+\cos\varphi_1, \sin\varphi_1\cos\varphi_2, \sin\varphi_1\sin\varphi_2).$$

On the other hand, using the elementary properties

$$\sin 2\alpha = 2\sin\alpha\cos\alpha, \quad 1+\cos 2\alpha = 2\cos^2\alpha,$$

one obtains

$$\int_{\mathcal{S}^2_+(u)} (e,u)\Phi(2e(e,u))\,de = \int_0^{\frac{\pi}{2}} d\varphi_1 \int_0^{2\pi} d\varphi_2 \sin\varphi_1\cos\varphi_1 \times$$
$$\Phi(2\cos^2\varphi_1, 2\cos\varphi_1\sin\varphi_1\cos\varphi_2, 2\cos\varphi_1\sin\varphi_1\sin\varphi_2)$$
$$= \int_0^{\frac{\pi}{2}} d\varphi_1 \int_0^{2\pi} d\varphi_2 \tfrac{1}{2}\sin 2\varphi_1 \times \qquad (1.22)$$
$$\Phi(1+\cos 2\varphi_1, \sin 2\varphi_1\cos\varphi_2, \sin 2\varphi_1\sin\varphi_2)$$
$$= \tfrac{1}{4}\int_0^\pi d\varphi_1 \int_0^{2\pi} d\varphi_2 \sin\varphi_1\Phi(1+\cos\varphi_1, \sin\varphi_1\cos\varphi_2, \sin\varphi_1\sin\varphi_2).$$

Comparing (1.21) and (1.22) gives (1.20).    ∎
**Proof of Theorem 1.2.**   Using Lemma 1.3 with

$$\Phi(z) = B(v,w,z-u)\left[f\left(v + \frac{|v-w|}{2}z\right)f\left(w - \frac{|v-w|}{2}z\right) - f(v)f(w)\right]$$

**Output:**

Note that

$$|2e(e,u) - u|^2 = 1 \quad \text{and} \quad |e+u|^2 = 2(1+(e,u)) = 2(e+u,u).$$

We prepare the proof by the following lemma.
Lemma 1.3. Let $\Phi$ be an appropriate test function and $u \in \mathcal{S}^2$. Then (cf. (1.17))

$$\int_{\mathcal{S}^2} \Phi(e+u)de = 4\int_{\mathcal{S}^2_+(u)}(e,u)\Phi(2e(e,u))de$$
$$= 4\int_{\mathcal{S}^2_+(u)}|(e,u)|\Phi(2e(e,u))de = 2\int_{\mathcal{S}^2}|(e,u)|\Phi(2e(e,u))de$$

Proof. Introducing spherical coordinates $\varphi_1 \in [0,\pi], \varphi_2 \in [0,2\pi]$ such that

$$e_1 = \cos\varphi_1, \quad e_2 = \sin\varphi_1\cos\varphi_2, \quad e_3 = \sin\varphi_1\sin\varphi_2$$

and $u = (1,0,0)$, one obtains

$$\int_{\mathcal{S}^2} \Phi(e+u)de =$$
$$\int_0^\pi d\varphi_1 \int_0^{2\pi} d\varphi_2 \sin\varphi_1\Phi(1+\cos\varphi_1, \sin\varphi_1\cos\varphi_2, \sin\varphi_1\sin\varphi_2).$$

On the other hand, using the elementary properties

$$\sin 2\alpha = 2\sin\alpha\cos\alpha, \quad 1+\cos 2\alpha = 2\cos^2\alpha,$$

one obtains

$$\int_{\mathcal{S}^2_+(u)}(e,u)\Phi(2e(e,u))de = \int_0^{\frac{\pi}{2}} d\varphi_1 \int_0^{2\pi} d\varphi_2 \sin\varphi_1\cos\varphi_1 \times$$
$$\Phi\left(2\cos^2\varphi_1, 2\cos\varphi_1\sin\varphi_1\cos\varphi_2, 2\cos\varphi_1\sin\varphi_1\sin\varphi_2\right)$$
$$= \int_0^{\frac{\pi}{2}} d\varphi_1 \int_0^{2\pi} d\varphi_2 \tfrac{1}{2}\sin 2\varphi_1 \times$$
$$\Phi(1+\cos 2\varphi_1, \sin 2\varphi_1\cos\varphi_2, \sin 2\varphi_1\sin\varphi_2)$$
$$= \tfrac{1}{4}\int_0^\pi d\varphi_1 \int_0^{2\pi} d\varphi_2 \sin\varphi_1\Phi(1+\cos\varphi_1, \sin\varphi_1\cos\varphi_2, \sin\varphi_1\sin\varphi_2).$$

Comparing (1.21) and (1.22) gives (1.20).
Proof of Theorem 1.2.    Using Lemma 1.3 with

$$\Phi(z) = B(v,w,z-u)\left[f\left(v + \frac{|v-w|}{2}z\right)f\left(w - \frac{|v-w|}{2}z\right) - f(v)f(w)\right]$$

| Method (test-dev set) | Backbone | Input size | AP | $AP_{50}$ | $AP_{75}$ | $AP_S$ | $AP_M$ | $AP_L$ |
|---|---|---|---|---|---|---|---|---|
| Center-guided(anchor-based): | | | | | | | | |
| Faster R-CNN [Ren et al., 2015] w/ FPN | ResNet-101 | $1000 \times 600$ | 36.2 | 59.1 | 39.0 | 18.2 | 39.0 | 48.2 |
| RetinaNet w/ FPN [Lin et al., 2017b] | ResNet-101 | $1333 \times 800$ | 40.8 | 61.1 | 44.1 | 24.1 | 44.2 | 51.2 |
| Cascade R-CNN [Cai and Vasconcelos, 2018] | ResNet-101 | $1333 \times 80$ | 42.8 | 62.1 | 46.3 | 23.7 | 45.5 | 55.2 |
| YOLOv4 [Bochkovskiy et al., 2020] | CSPDarkNet-53 | $608 \times 608$ | 43.5 | 65.7 | 47.3 | 26.7 | 46.7 | 53.3 |
| Center-guided (anchor-free): | | | | | | | | |
| CenterNet [Zhou et al., 2019] | Hourglass-104 | $512 \times 512$ | 42.1 | 61.1 | 45.9 | 24.1 | 45.5 | 52.8 |
| FCOS [Tian et al., 2019] | ResNeXt-101 | $1333 \times 800$ | 42.1 | 62.1 | 45.2 | 25.6 | 44.9 | 52.0 |
| Reppoints [Yang et al., 2019] | ResNet-101 | $1333 \times 806$ | 45.0 | 66.1 | 49.0 | 26.6 | 48.6 | 57.5 |
| Corner-guided: | | | | | | | | |
| CenterNet [Duan et al., 2019] | Hourglass-104 | $511 \times 511$ | 44.9 | 62.4 | 48.1 | 25.6 | 47.4 | 57.4 |
| CentripetalNet [Dong et al., 2020] | Hourglass-104 | $511 \times 511$ | 45.8 | 63.0 | 49.3 | 25.0 | 48.2 | 58.7 |
| CPN (two-stage) [Duan et al., 2020] | Hourglass-104 | $511 \times 111$ | 47.0 | 65.0 | 51.0 | 26.5 | 50.2 | 60.7 |
| CornerNet baseline [Law and Deng, 2018] | Hourglass-104 | $511 \times 511$ | 40.5 | 56.5 | 43.1 | 19.4 | 42.7 | 53.9 |
| w/ Corner Affinity | Hourglass-104 | $511 \times 511$ | 46.3 | 64.0 | 49.9 | 27.4 | 49.3 | 58.7 |
| Improvement | - | - | ↑5.8 | ↑7.5 | ↑6.8 | ↑8.0 | ↑6.6 | ↑4.8 |
| Method (val set) | Backbone | Feature | AP | $AP_{50}$ | $AP_{75}$ | $AP_S$ | $AP_M$ | $AP_L$ |
| DETR [Carion et al., 2020] | ResNet-50 | Encoder | 43.3 | 63.1 | 45.9 | 22.5 | 47.3 | 61.1 |
| DETR [Carion et al., 2020] | ResNet-101 | Encoder | 44.9 | 64.7 | 47.7 | 23.7 | 49.5 | 62.3 |
| Sparse R-CNN [Sun et al., 2021] | ResNet-50 | FPN | 42.8 | 61.2 | 45.7 | 26.7 | 44.6 | 57.6 |
| Sparse R-CNN [Sun et al., 2021] | ResNet-101 | FPN | 44.1 | 62.1 | 47.2 | 26.1 | 46.3 | 59.7 |
| w/ Corner Affinity | Hourglass-52 | Single | 43.2 | 60.8 | 46.1 | 25.1 | 46.8 | 57.7 |
| w/ Corner Affinity | Hourglass-104 | Single | 45.1 | 62.9 | 48.3 | 26.7 | 48.5 | 59.8 |

$$d_L(C_L, \{v'\}) = |C_L| + |\{v'\}| + 2(d_T(C_L, v') - 1)$$
$$= |C_v| - 1 + |S_v^*| + 2(\text{rad}\,T - 1)$$
$$= |C_v| + |S_v^*| + 2(d_T(C_v, S_v^*) - 1)$$
$$\quad - 1 + 2(\text{rad}\,T - d_T(C_v, S_v^*))$$
$$= d_L(C_v, S_v^*) + 1 + 2(\text{rad}\,T - 1 - d_T(C_v, S_v^*))$$
$$= e_L(C_v) + 1 + 2(\text{rad}\,T - 1 - d_T(C_v, S_v^*)).$$

Figure 4: The formatted text OCR ability of GOT. GOT works well on full-page texts and table/formula slice texts. These input forms are the most commonly used in document OCR, which proves that GOT has great prospects in application.

**Prompt: OCR:**

[21], and GuidedBackpropagation [22]) to explain image captioning predictions with respect to the image content and the words of the sentence generated so far. These approaches provide high-resolution image explanations for CNN models [22], [23]. LRP also provides plausible explanations for LSTM architectures [24], [25]. Figure 1 shows an example of the explanation results of attention-guided image captioning models. Taking LRP as an example, both positive and negative evidence is shown in two aspects: 1) for image explanations, the contribution of the image input is visualized as heatmaps; 2) for linguistic explanations, the contribution of the previously generated words to the latest predicted word is shown.

The explanation results in Figure 1 exhibit intuitive correspondence of the explained word to the image content and the related sequential input. However, to our best knowledge, few works quantitatively analyze how accurate the image explanations are grounded to the relevant image content and whether the highlighted inputs are used as evidence by the model to make decisions. We study the two questions by quantifying the grounding property of attention and explanation methods and by designing an ablation experiment for both the image explanations and linguistic explanations. We will demonstrate that explanation methods can generate image explanations with accurate spatial grounding property, meanwhile, reveal more related inputs (pixels of the image input and words of the linguistic sequence input) that are used as evidence for the model decisions. Also, explanation methods can disentangle the contributions of the image and text inputs and provide more interpretable information than purely image-centered attention.

With explanation methods [26], we have a deeper understanding of image captioning models beyond visualizing the attention. We also observe that image captioning models sometimes hallucinate words from the learned sentence correlations without looking at the images and sometimes use irrelevant evidence to make predictions. The hallucination problem is also discussed in [27], where the authors state that it is possibly caused by language priors or visual mis-classification, which could be partially due to the biases present in the dataset. The image captioning models tend to generate those words and sentence patterns that appear more frequently during training. The language priors are helpful, though, in some cases. [28] incorporates the inductive bias of natural language with scene graphs to facilitate image captioning. However, language bias is not always correct, for example, not only men ride snowboards [29] and bananas are not always yellow [30], [31]. To this end, [29] and [31] attempted to generate more grounded captions by guiding the model to make the right decisions using the right reasons. They adopted additional annotations, such as the instance segmentation annotation and the human-annotated rank of the relevant image patches, to design new losses for training.

In this paper, we reduce object hallucination by a simple *LRP-inference fine-tuning* (LRP-IFT) strategy, without any additional annotations. We firstly show that the explanations, especially LRP, can weakly differentiate the grounded (true-positive) and hallucinated (false-positive) words. Secondly, based on the findings that LRP reveals the related features of the explained words and that the sign of its relevance scores indicates supporting versus opposing evidence (as shown in Figure 1), we utilize LRP explanations to design a re-weighting mechanism for the context representation. During fine-tuning, we up-scale the supporting features and down-scale the opposing ones using a weight calculated from LRP relevance scores. Finally, we use the re-weighted context representation to predict the next word for fine-tuning.

LRP-IFT is different from standard fine-tuning which weights the gradients of parameters with small learning rates to gradually adapt the model parameters. Instead, it pinpoints the related features/evidence for a decision and guides the model to tune more on those related features. This fine-tuning strategy resembles how we correct our cognition bias. For example, when we see a green banana, we will update the color feature of bananas and keep the other features such as the shape.

We will demonstrate that LRP-IFT can help to de-bias image captioning models from frequently occurring object words. Though language bias is intrinsic, we can guide the model to be more precise when generating frequent object words rather than hallucinate them. We implement the LRP-IFT on top of pre-trained image captioning models trained with Flickr30K [32] and MSCOCO2017 [33] datasets and effectively improve the mean average precision (mAP) of predicted frequent object words evaluated across the test set. At the same time, the overall performance in terms of sentence-level evaluation metrics is maintained.

The contributions of this paper are as follows:

- We establish explanation methods that disentangle the contributions of the image and text inputs and explain image captioning models beyond visualizing attention.
- We quantitatively measure and compare the properties of explanation methods and attention mechanisms, including tasks of finding the related features/evidence for model decisions, grounding to image content, and the capability of debugging the models (in terms of providing possible reasons for object hallucination and differentiating hallucinated words).
- We propose an LRP-inference fine-tuning strategy that reduces object hallucination and guides the models to be more precise and grounded on image evidence when predicting frequent object words. Our proposed fine-tuning strategy requires no additional annotations and successfully improves the mean average precision of predicted frequent object words.

In the rest of this paper, Section II introduces recent image captioning models, the state-of-the-art explanation methods for neural networks, and other related works. In Section III, we will introduce the image captioning model structures applied in this paper. The adaptations of explanation methods to attention-guided image captioning models are summarized in Section IV. The analyses of attention and explanations and our proposed LRP-inference fine-tuning strategy are introduced in Section V.

**Output:**

[21], and GuidedBackpropagation [22]) to explain image captioning predictions with respect to the image content and the words of the sentence generated so far. These approaches provide high-resolution image explanations for CNN models [22], [23], LRP also provides plausible explanations for LSTM architectures [24], [25]. Figure 1 shows an example of the explanation results of attention-guided image captioning models. Taking LRP as an example, both positive and negative evidence is shown in two aspects: 1) for image explanations, the contribution of the image input is visualized as heatmaps; 2) for linguistic explanations, the contribution of the previously generated words to the latest predicted word is shown.

The explanation results in Figure 1 exhibit intuitive correspondence of the explained word to the image content and the related sequential input. However, to our best knowledge, few-works quantitatively analyze how accurate the image explanations are grounded to the relevant image content and whether the highlighted inputs are used as evidence by the model to make decisions. We study the two questions by quantifying the grounding property of attention and explanation methods and by designing an ablation experiment for both the image explanations and linguistic explanations. We will demonstrate that explanation methods can generate image explanations with accurate spatial grounding property, meanwhile, reveal more related inputs (pixels of the image input and words of the linguistic sequence input) that are used as evidence for the model decisions. Also, explanation methods can disentangle the contributions of the image and text inputs and provide more interpretable information than purely image-centered attention.

With explanation methods [26], we have a deeper understanding of image captioning models beyond visualizing the attention. We also observe that image captioning models sometimes hallucinate words from the learned sentence correlations without looking at the images and sometimes use irrelevant evidence to make predictions. The hallucination problem is also discussed in [27], where the authors state that it is possibly caused by language priors or visual mis-classification, which could be partially due to the biases present in the dataset. The image captioning models tend to generate those words and sentence patterns that appear more frequently during training. The language priors are helpful, though, in some cases. [28] incorporates the inductive bias of natural language with scene graphs to facilitate image captioning. However, language bias is not always correct, for example, not only men ride snowboards [29] and bananas are not always yellow [30], [31]. To this end, [29] and [31] attempted to generate more grounded captions by guiding the model to make the right decisions using the right reasons. They adopted additional annotations, such as the instance segmentation annotation and the human-annotated rank of the relevant image patches, to design new losses for training.

In this paper, we reduce object hallucination by a simple LRP-inference fine-tuning (LRP-IFT) strategy, without any additional annotations .We firstly show that the explanations, especially LRP, can weakly differentiate the grounded (true-positive) and hallucinated (false-positive) words. Secondly, based on the findings that LRP reveals the related features of the explained words and that the sign of its relevance scores indicates supporting versus opposing evidence (as shown in Figure 1), we utilize LRP explanations to design a re-weighting mechanism for the context representation. During fine-tuning, we up-scale the supporting features and down-scale the opposing ones using a weight calculated from LRP relevance scores. Finally, we use the re-weighted context representation to predict the next word for fine-tuning.

LRP-IFT is different from standard fine-tuning which weights the gradients of parameters with small learning rates to gradually adapt the model parameters. Instead, it pinpoints the related features/evidence for a decision and guides the model to tune more on those related features .This fine-tuning strategy resembles how we correct our cognition bias. For example, when we see a green banana, we will update the color feature of bananas and keep the other features such as the shape.

We will demonstrate that LRP-IFT can help to de-bias image captioning models from frequently occurring object words. Though language bias is intrinsic, we can guide the model to be more precise when generating frequent object words rather than hallucinate them. We implement the LRP-IFT on top of pre-trained image captioning models trained with Flickr30K [32] and MSCOCO2017 [33] datasets and effectively improve the mean average precision (mAP) of predicted frequent object words evaluated across the test set. At the same time, the overall performance in terms of sentence-level evaluation metrics is maintained.

The contributions of this paper are as follows:
• We establish explanation methods that disentangle the contributions of the image and text inputs and explain image captioning models beyond visualizing attention.
• We quantitatively measure and compare the properties of explanation methods and attention mechanisms, including tasks of finding the related features/evidence for model decisions, grounding to image content, and the capability of debugging the models (in terms of providing possible reasons for object hallucination and differentiating hallucinated words).
• We propose an LRP-inference fine-tuning strategy that reduces object hallucination and guides the models to be more precise and grounded on image evidence when predicting frequent object words .Our proposed fine-tuning strategy requires no additional annotations and successfully improves the mean average precision of predicted frequent object words.

In the rest of this paper, Section II introduces recent image captioning models, the state-of-the-art explanation methods for neural networks, and other related works. In Section III, we will introduce the image captioning model structures applied in this paper. The adaptations of explanation methods to attention-guided image captioning models are summarized in Section IV. The analyses of attention and explanations and our proposed LRP-inference fine-tuning strategy are introduced in Section V.

Figure 5: The plain text (document) OCR ability of GOT. For double-column documents with high text density, GOT can still handle them well, proving the excellent text perception ability.

**Prompt: OCR** **Output:**

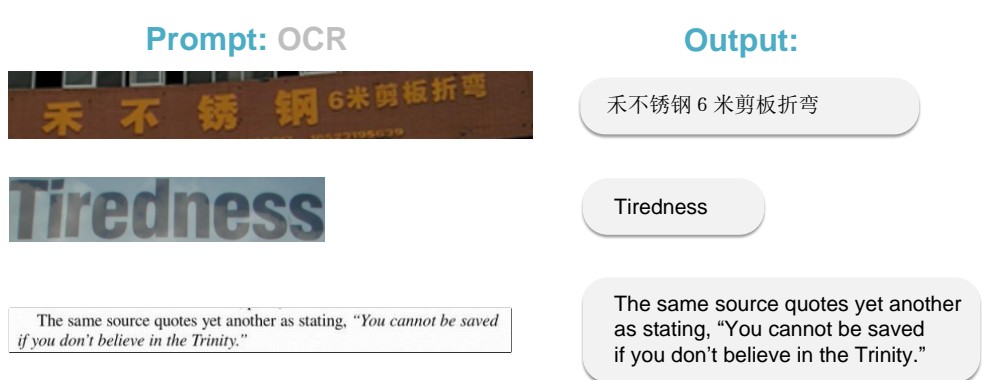

禾不锈钢 6 米剪板折弯

Tiredness

The same source quotes yet another as stating, "You cannot be saved if you don't believe in the Trinity."

**Prompt: OCR/ [green]OCR/ OCR with format:**

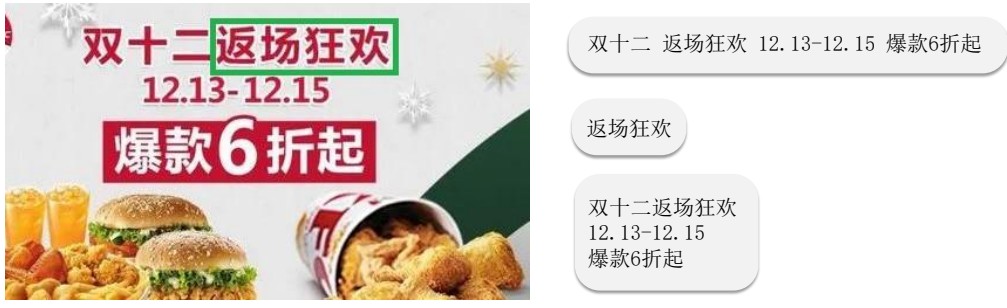

双十二 返场狂欢 12.13-12.15 爆款6折起

返场狂欢

双十二返场狂欢
12.13-12.15
爆款6折起

**Prompt: [x1,y1,x2,y2] OCR with format:**

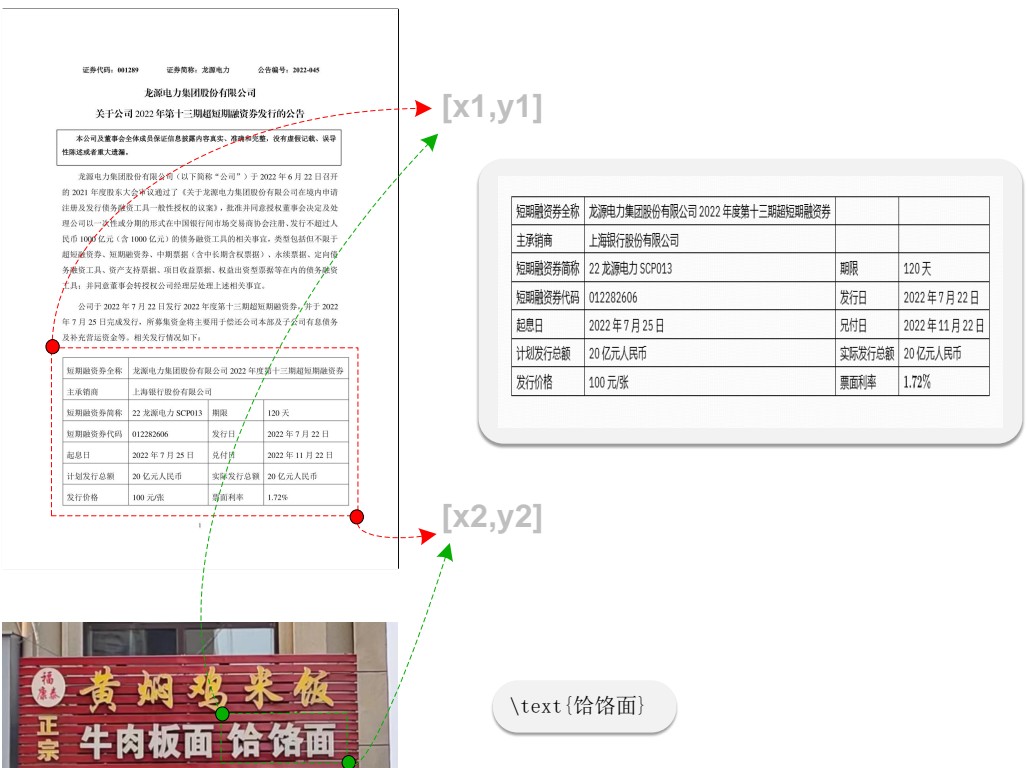

\text{饸饹面}

Figure 6: Scene OCR and fine-grained OCR results of GOT. We equip GOT with more interactive fine-grained OCR tasks, allowing it to output OCR results of regions of interest based on prompts.

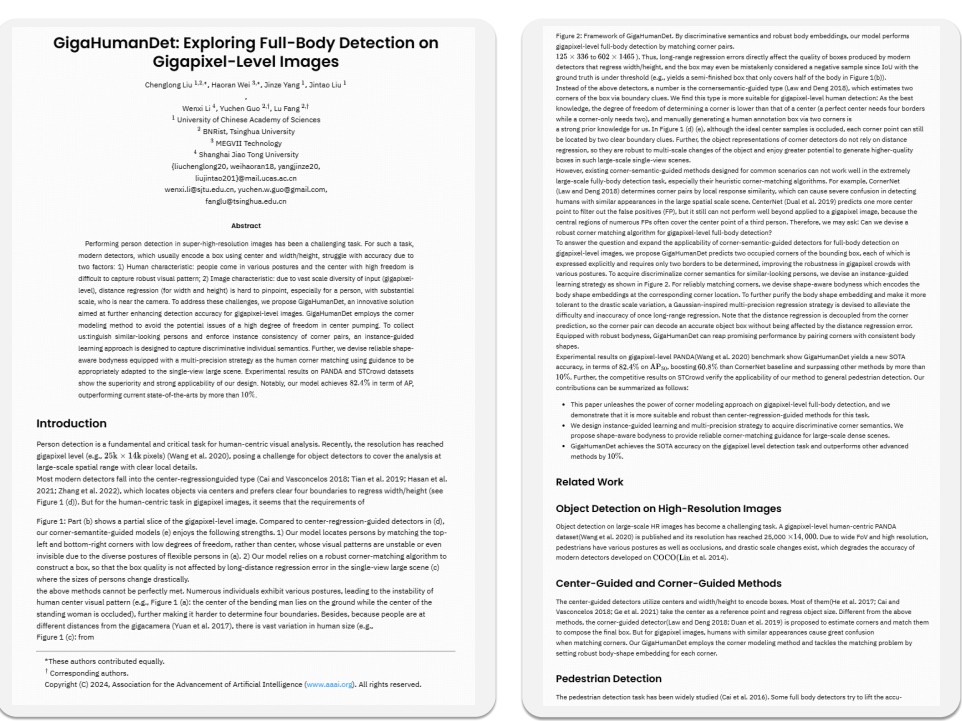

Figure 7: Dynamic resolution of GOT for high-resolution images. In the dual-page paper reading mode shown in the figure (data is from Liu et al. (2024b)), the input resolution of the original GOT is not sufficient to handle it. Therefore, we adapt dynamic resolution technology to make the model no longer limited to the size of the image.

**Prompt:** OCR with format across multi pages:

**Output:**

Figure 8: Multi-page (document) OCR ability of GOT. With this feature, researchers can continue to train the GOT with multi-page PDF-text pairs, such as Arxiv paper with *.tex* file.

**Prompt:** OCR with format:                **Output:**

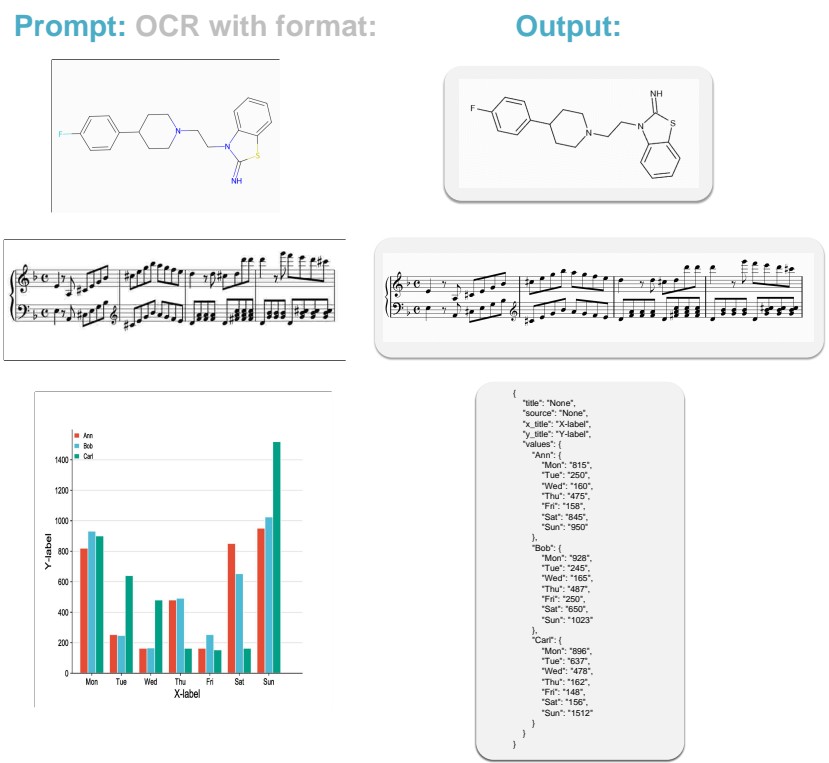

Figure 9: More general OCR results. GOT can process molecular formulas, sheet music, and charts.

**Prompt:** OCR with format:  **Output:**

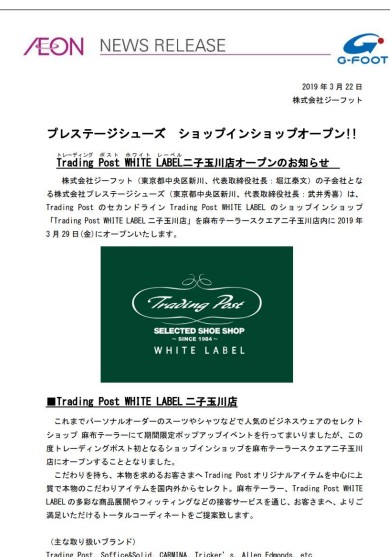

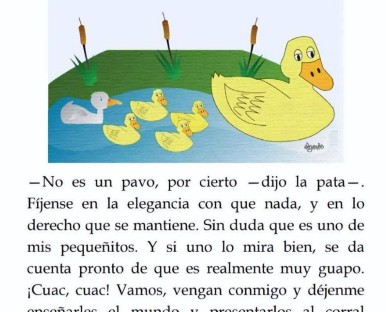

Figure 10: We do not specifically introduce additional OCR capabilities for GOT other than Chinese and English. Yet the PDF data we crawled may contain a small amount of text in other languages, leading to the GOT seeming to have the ability to recognize other languages. However, we cannot guarantee the OCR quality of other languages. Therefore, we recommend fine-tuning the model with corresponding data if this feature is needed.

