# OpenReview forum: "General OCR Theory:  Towards OCR-2.0 via a Unified End-to-end Model"
_ICLR.cc/2025/Conference — ICLR 2025 Conference Withdrawn Submission_

### Official Review · Reviewer_rYsc · 2024-10-30

**Soundness:** 2
**Presentation:** 3
**Contribution:** 2
**Rating:** 5
**Confidence:** 4

**Summary:**

This paper describes a single unified framework to perform end-to-end OCR in different kinds of images (documents, scene images, handwritten text, charts, music sheets, math formula). The framework relies on collecting a large amount of data for every type of image, partially from public data sources, partially automatically rendered. Then, a curriculum strategy is employed to train the model based on standard encoder and decoder architectures. In a first stage, only a limited number of OCR tasks with limited variability are used to train the encoder using a simple decoder and, progressively more data, tasks and the final decoder architecture are included in subsequent training stages. Experimental results compare the proposed approach with other generic models based on multimodal LLMs

**Strengths:**

- Compared to other unified end-to-end frameworks for multi-task OCR based on multimodal LLMs, the proposed approach is efficient and the model is relatively smail.
- The proposal of a new training strategy adding complexity increasingly to the model, either from the point of view of the model and the data used for training.
- The generation of a large collection of data to train the model can be useful for advancing research in generic OCR (if the data is made public after publication)

**Weaknesses:**

- The paper lacks contextualization and comparison with previous SoA OCR methods not based on LLMs, specialized on each of the individual OCR tasks. Related work lacks a much better discusion and reference to all existing specific methods for text recognition in different tasks (scene text, documents, handwritten text, ...). In the experimental results I also miss comparison with specific OCR methods in each task, even in some tasks comparison with existing commerical OCR tools.
- Following the previous comment, I think that the papser should also use common standard benchmarks and datasets in some specific OCR tasks. In the past years there has been a huge effort in the text recognition community to create standard benchmarks for evaluation, that are ignored in the paper. Using these common benchmarks (for all the tasks where this is possible) would help to get a better understanding of the contribution of the proposed approach in comparison with existing OCR techniques.
- As far as I understand, most of the images used to train and evaluate the proposed approach are very clean images, collected from clean pdf documents or automatically rendered, without the kind of noise, distortion, low resolution problems, ... that can be encountered when dealing with real images.
- I miss some analysis of the contribution of each of the training stages in the final performance of the model.

**Questions:**

- Some more details would be necesary on how metrics are computed given the full recognized text and ground-truth .
- Also some more details on how the OCR task on charts is defined

---

> ### Author Response · Authors · 2024-11-15
> **Response to Reviewer rYsc**
>
> **Thank you very much for your review. We will do our best to address your concerns:**
> - After a year of rapid development, there is already a consensus that LVLMs have higher ceilings and potential for OCR compared to traditional OCR models (including traditional SOTA models). Therefore, in the limited paper space available, our experiments mainly focus on comparing the most advanced LVLMs. After the paper is made public, we are happy to conduct more comparisons and strive to promote open-source efforts to allow more people to evaluate the model’s performance across various subtasks.
>
>   Additionally, it is important to note that this is a research paper, and we believe comparing commercial tools is inappropriate. Second, the model algorithm proposed in this paper should serve as a baseline for OCR-2.0 technology, and we hope this paper brings new research perspectives to the OCR field. GOT is not a tool, nor is it called “OCR anything,” so we cannot cover massive domain-specific tests, and excessive demands would be unreasonable. For the related work section, we will include more traditional specialized-task OCR models in the next version of the manuscript.  Thank you.
>
> - Thanks for your comments. The reason we did not test on the common standard datasets (natural images) is that our primary comparison targets are LVLMs, and we are unsure whether they have used the relevant data for training. Our natural image test data is sourced from the same origins as most LVLMs (such as Laion, Wukong, etc.), and this choice was made to ensure a fair comparison.
>
>   Besides, we acknowledge the efforts and importance of previous OCR-related works and benchmarks, but we also believe that benchmarks will continue to change with the development of OCR technology. One thing that also must be acknowledged is that the classic benchmarks from the past are no longer suitable for evaluating the OCR capabilities of LVLMs, because previous benchmarks were designed for those OCR models with detection-cropping processes, and many images were in small cropped format. When we have OCR-2.0 models that can perceive text in entire images, we need to consider whether we should rigidly continue using previous benchmarks. We hope that the proposed new benchmark will also be one of the contributions of our paper.
>
> - Our training is designed to save resources under large models and long token lengths, rather than to enhance performance. For example, in Stage 1, we use a smaller language model as the decoder to quickly warm up the encoder, which allows us to save about half of the resources. For excessively long texts, such as multi-page content or scenarios involving extremely long image crops, we place them in Stage 3, where we freeze the encoder to reduce the additional GPU memory costs associated with encoder activations.
>
>   In summary, the value of GOT's training paradigm cannot be judged by previous approaches because earlier models did not handle extremely long tokens and did not have large parameter counts. The resource bottleneck problems they encountered were much smaller in scale.
>
> **[Question]**
> Our evaluation metrics are common standards, and due to space limitations, we did not include them in the main text. The calculation methods for the chart metrics are referenced in our paper, such as OneChart (Chen et al.).
>
> **Sincerely hope that my response has resolved your concerns.**

---

> > ### Comment · Reviewer_rYsc · 2024-11-19
> >
> > Thank you for you response. I still have some comments for further discussion:
> >
> > - It's true that LVLMs can have the potential to perform general OCR tasks, but I think that comparison with SoA task-specific OCR methods is still relevant to get a better insight of the pros and cons of a generic LVLM versus specific models, even if results could show superior performance of LVLMs.
> > - It's also true that LVLMs can have seen the test images of standard benchmarks in their training. But still, I think that using those benchmarks is still relevant since they are the best way to compare with previous work, even if that implies being cautious about results obtained by LVLMs. And there are several previous benchmarks (icdar-2015, coco-text, total-tex, ctw500, ...)  that also consider end-to-end text recognition, comprising detection and recognition.
> > - Even if the three-stage strategy is not designed to improve performance, some evaluation of the need to have each of the three stages would be useful to assess the whole pipeline.
> > - Although you use standard metrics, I still think that their application to end-to-end ocr evaluation, where two sets of words have to be compared and matched, would deserve some more details. In addition, while for plain text recognition you use word-level segmentation, for scene text you use character-level segmentation. How does this affect to the way that the metrics are computed?
> > - For the OCR task on charts, the reference to the OneChart paper is useful, but I realize that the results of OneChart are not included in table, and they are, in some cases, better than the results reported by your method, specially on the PlotQA dataset.

---

### Official Review · Reviewer_bE94 · 2024-11-01

**Soundness:** 3
**Presentation:** 2
**Contribution:** 4
**Rating:** 6
**Confidence:** 5

**Summary:**

The paper introduces the GOT model (General OCR Theory), to improve upon traditional OCR systems (OCR-1.0). With 580 million parameters, GOT processes various artificial optical signals and supports multiple output formats. It features interactive region-level recognition and incorporates dynamic resolution and multipage OCR, showing superior performance in experiments.

**Strengths:**

1. This article presents a unified approach to OCR recognition tasks, making it one of the most comprehensive OCR models to date with sufficient tasks.
2. The proposed GOT method employs a three-stage pre-training and fine-tuning process to achieve the experimental results outlined in the paper.
3. The GOT method addresses various OCR recognition problems across multiple scenarios (natural scenes, documents, etc.), as well as different levels of granularity, such as document-level and region-level recognition.
4. Multiple datasets are constructed to conduct these diverse settings of these OCR recognition tasks.

**Weaknesses:**

1. The writing of this article needs further improvement, as several key details are missing. For example, when discussing the method, it is unclear how to distinguish between different tasks. Does it involve using a question as input to the decoder, similar to existing MLLMs?
2. In the experiment, the paper does not conduct the comparisons on the benchmarks of  OCRBench, InfoVQA, and DocVQA. Is this because the proposed method does not support QA? (You did not clarify how you distinguish between different tasks?)
3. This paper mainly focuses on recognition issues related to OCR tasks and does not address detection problems. One possible reason could be that both the current encoder-decoder and decoder-only architectures struggle with coordinate regression prediction, which may have prevented you from tackling detection tasks.
4. Additionally, there is a lack of comparison with methods like Kosmos?

**Questions:**

Show in the part of Weakness.

---

> ### Author Response · Authors · 2024-11-15
> **Response to Reviewer bE94**
>
> **Thank you very much for your comments. We will do our best to address your concerns:**
> - For different tasks, routing through different prompts is necessary, which can be found in the paper and illustrated in the supplementary materials. More details can be found in the submitted codes.  In the future,  we will actively promote open-source to make the usage of GOT clearer.
> - We would like to politely point out that the tasks you mentioned are not OCR tasks; they fall under VQA (Visual Question Answering) tasks. Even if they involve texts, text-driven VQA should not be regarded as  OCR, i.e.,  high scores in VQA do not directly reflect a model’s OCR capabilities. These tasks are merely downstream applications of OCR, and OCR-2.0 is not designed for downstream tasks.
>
>   For example, in DocVQA samples, OCR capabilities would involve transcribing the entire document in its original format, whereas QA does not necessarily require a strong perception of the entire text. In QA, learning certain answering patterns may allow the model to find correct answers by searching through parts of the text. Therefore, even text-dependent QA and OCR are not equivalent.  We should not include these QA metrics. Thanks.
>
> - The “R” in OCR stands for Recognition and should not be tied to detection. Previous methods required detection because they lacked strong perception and full-text recognition capabilities, and thus relied on detection boxes to crop specific areas for recognition. A key point of OCR 2.0 is the ability to perceive and recognize dense, full-page images without relying on detection. Regarding your concerns about the detection capabilities of auto-regression models, although it is unrelated to our model, we are willing to discuss this with you:
>
>   We believe that there is an essential difference between the current LVLM paradigm and the original detection models for detection tasks. The LVLM’s approach of using next-token predictions to output text (string) for predicting numerical coordinates (float) is clearly at a disadvantage under the current training and testing paradigms. However, this does not mean that this architecture cannot perform well on this task. Referring to the process of human-annotated detection data, the current LVLM architecture should perform multiple rounds of visual inspection and box refinement for a target to achieve good results, which implies multiple times perceptions. This point is also applicable to the aforementioned VQA tasks. The current LVLM architecture is best suited for quick recognition, which is also the original intention behind promoting OCR 2.0.
>
> **Sincerely hope that my response has resolved your concerns.**

---

### Official Review · Reviewer_m6tW · 2024-11-01

**Soundness:** 3
**Presentation:** 4
**Contribution:** 3
**Rating:** 5
**Confidence:** 5

**Summary:**

This paper introduces an so-called OCR-2.0 model named GOT, designed for advanced Optical Character Recognition tasks. It proposes a new OCR model, emphasizing end-to-end architecture, low training costs, and versatility in recognizing a wide range of artificial optical characters. The model, with 580M parameters, incorporates a high-compression encoder and a long-context decoder for handling various OCR tasks.

GOT is evaluated on multiple OCR tasks, demonstrating superior performance in plain document OCR, scene text OCR, formatted document OCR, fine-grained OCR, and more general OCR tasks like sheet music, geometric shapes, and chart OCR.

**Strengths:**

- The paper introduces a unified OCR-2.0  model, emphasizing an end-to-end architecture whichl is designed to handle various OCR tasks efficiently.

- GOT demonstrates versatility by recognizing a wide range of artificial optical characters, including sheet music, geometric shapes, and charts. The model can adapt to different input styles and output formats, enhancing readability for formulas and tables.

- This paper is well written and well organized.

- The idea of OCR 2.0 is interesting and novel.

**Weaknesses:**

(1) The term "general OCR theory" is not appropriate as the paper does not present any rigorous theory. It is suggested to consider alternative terms such as General OCR Technology/Framework/Pipeline/Methodology.

(2) Dataset construction is a significant contribution of this work. The authors utilized data engineering methods to create a substantial amount of non-public training data. If these datasets are not made publicly accessible, it will make it challenging for other researchers to perform fair comparisons under the same settings as this paper.

(3). In section 4.2.2, the authors collected 400 natural scene text images for testing. Why did they not use publicly available datasets in this domain (such as CTW1500, ReCTS, etc.) to evaluate the performance of GOT on natural scene text?  I am wondering if the proposed method on these public datasets can achieve state-of-the-art (SOTA) performance.

**Questions:**

1. How does the performance of the proposed method fare on openly and widely used page-level datasets (such as CASIA-HWDB, HCCDoc, CTW1500, ReCTS, IAM, CROHME16/19, etc.)? Why was the effectiveness of the proposed method not tested on these commonly used datasets in the community?

2. Are the test datasets used in sections 4.2.2, 4.2.3, and 4.2.5 open-source?

3. In the references, proprietary acronyms should be capitalized, for example, CASIA, IAM, HDWB, BLIP, etc.

Additional comment:  I do not agree that low training and inference costs must be a characteristic of OCR 2.0. As a new technology framework or paradigm for OCR in the era of AGI, it should also possess scalability capabilities.

---

> ### Author Response · Authors · 2024-11-15
> **Response to Reviewer m6tW**
>
> **Thank you very much for your professional suggestions. We will do our best to address your concerns:**
>
> **[Weakness]**
> - Thank you for the good suggestion. Changing “theory” to “technology” is a good idea. It will not alter the model’s abbreviation, GOT. We will consider your suggestion in the next manuscript. Thank you once again!
> - After discussion, we have reached a consensus that after GOT is successfully published, we will open-source most of the training data to further promote community development. We maintain a very open attitude towards open-source and will continue to optimize this GOT open-source-plan in the future, enabling everyone to better understand and reproduce the model.
> - The reason we did not test on the dataset you mentioned is that our primary comparison targets are LVLMs, and we are unsure whether they have used the relevant data for training. Our natural image test data is sourced from the same origins as most LVLMs's OCR data (such as Laion, Wukong, etc.), and this choice was made to ensure a fair comparison.
>
> Regarding the dataset you mentioned, most of them were not used during our training. One reason is the relatively small data volume, and another is that the annotation formats of each dataset differ. We were concerned that training them might overfit specific, non-generalizable annotations, which could affect the model’s data scaling-up ability. Fortunately, we can perform separate SFT for each dataset. We selected CTW1500 and ReCTS, and fine-tuned for just one epoch. Because GOT did not have detection capabilities, we split the output results by spaces and used the ‘include’ method to determine accuracy by checking if the output is in the ground truth. GOT achieved 83.9% and 84.7% accuracy on both CTW1500 and ReCTS, which I provide as a reference for you.
>
> **[Question]**
> - As mentioned above, the reason we did not evaluate certain public datasets is that we were uncertain whether other LVLMs had used them, and since each dataset has specific annotation formats, zero-shot evaluation was not advisable. To ensure a fair comparison, we also refrained from including most of these datasets in our training. A feasible approach is that we will include a comparative analysis of the supervised fine-tuning (SFT) performance of various LVLMs across the mentioned datasets in our future manuscript.
> - Yes, all test datasets will be open-sourced. If the paper goes smoothly, we will also make our training data publicly available in the future.
> - Thank you for your suggestions. We will correct some typos in our future manuscript.
> - I strongly agree with the latter point of your suggestion about OCR-2.0 - high extensibility. Currently, we have successfully fine-tuned GOT across various languages, such as Arabic, Haitian, and Indian languages, which demonstrate extremely strong extensibility. Thus, we believe GOT will be an excellent baseline.
>
> **Sincerely hope that the response has resolved your concerns.**

---

> > ### Comment · Reviewer_m6tW · 2024-11-23
> > **Insufficient Experimental Validation and Comparative Analysis for Proposed OCR 2.0 Framework**
> >
> > While the authors have addressed some of my concerns, several critical issues remain inadequately addressed.
> >
> > First, regarding the experimental comparisons: If the work primarily focuses on LVLMs, it should follow standard practices in the field by conducting comprehensive evaluations on established benchmarks such as OCRBench, TextVQA, DocVQA and so on, comparing against state-of-the-art methods like Monkey, MiniMonkey, QWen2-VL and more. This concern has also been raised by other reviewers. The differences in data formats should not be used as a justification for omitting these essential comparisons.
> >
> > Second, if the authors claim this work represents an "OCR 2.0" framework, it should at minimum demonstrate competent performance on traditional OCR datasets (such as IAM, CROHME, CTW1500, etc.) that are well-handled by existing OCR 1.0 methods. While the authors provided some preliminary results on CTW1500 and ReCTS after fine-tuning, a more comprehensive evaluation is needed to demonstrate the framework's generalizability and extensibility.
> >
> > Based on the current manuscript and reported experimental results, I remain unconvinced that the proposed method represents a breakthrough technology or effectively addresses real-world application challenges. While the authors show some promising results on specific datasets, the lack of systematic comparison with existing methods and limited evaluation scope make it difficult to fully assess the method's capabilities and advantages over current approaches.
> >
> > I acknowledge the authors' commitment to open-sourcing their data.  I hope they will keep their promises if this paper will be published somewhere.

---

### Official Review · Reviewer_Qy2H · 2024-11-02

**Soundness:** 3
**Presentation:** 1
**Contribution:** 3
**Rating:** 3
**Confidence:** 4

**Summary:**

The manuscript proposes a unified end-to-end 2.0 model for OCR, called GOT (General OCR Theory) using LVLMs (Large Vision Language Models). The architecture contains 80M parameters in the encoder, and 500M parameters in the decoder tackling long-contexts. Region-based recognition, dynamic resolutions and multi-page OCR are few other properties of GOT. It supports English and Chinese and can produce structured formats like markdown, tikz, smiles and kern.

GOT has a 3-stage training process: pre-training the vision encoder, joint-training of encoder and decoder, and finally the post-training of the language decoder. The performance is compared against SOTA methods on various scores like edit distance, F1, BLEU and METEOR, and seems to out-perform against majority of the SOTA methods. The results on markdown, sheet music, geometry and number-centric charts are also presented.

**Strengths:**

The paper presents a unified end-to-end model for a gamut of OCR documents, including sheet music, geometry and number-centric charts. It replaces the cascaded OCRs specialized in different document types.

The way the three stages of training are applied to unify a diverse set of OCR tasks (scene, document, chart, music sheets, etc.) within a single OCR is interesting. The task-oriented fine-tuning is limited to post-processing the language decoder. Freezing the vision encoder avoids increasing the computational demands and ensures foundational visual understanding is stable across the tasks.

The results are compared against the SOTA methods on a variety of metrics including F1-scores, edit distances, BLEU and METEOR values, and seem to outperform majority of the methods. For box-guided and color-guided OCR, specific comparison to Fox Lie et al. seems to outperform against all the metrics.

**Weaknesses:**

The weakness of the paper lies in its novelty. The 3-stage training process is well known in the literature. For example, many existing frameworks in OCR, vision-language and LVLMs decouple encoder pre-training from the rest of the pipeline. The vision encoders are usually pre-trained on a wide variety of data to create a foundational understanding of text and scene. The joint training of vision and language pieces is again known in models in UniT, BLIP and LVLMs. Lastly, the fine-tuning of the language decoder piece is again seen in T5 etc. Perhaps, the prime novelty is the application of these methods to an OCR problem, smarts about synthetic data generation and OCR-specific fine-tuning.

The other weakness of the paper is in its presentation. The paper is overall hard to follow, as it continues to mix, architecture, training, data and task-specific details all together, and does not lay out in separate sub-sections. E.g. the section 3.2.1 starts with the architecture, dives into input sizes, parameter sizes, goes through data peculiarities (natural scenes, cropped slices) and training process all in one paragraph. A lot of architecture diagrams can be added to aid the reading.

Lastly, in experiments, ablation studies are missing to underscore the importance of each of the stages, data types. Latency studies, comparisons with SOTA methods, and failure cases are missing.

**Questions:**

1. Section 4.1 lists joint training and post-training for only 1 epoch. Usually multiple epochs are required to train a model. While post-training can be understood as vision encoder and much of language decoder may already be well-trained from prior stages, 1 epoch for joint training seems pretty small. Any reason why that worked? Is there a study on how more epochs affected the outcome? Is it possible that there isn't much data diversity between training and test set, and hence, 1 epoch is enough?

2. What are the training/inference latency gains by using a smaller size model like GOT compared to Qwen-VL-MAX or others?

---

> ### Author Response · Authors · 2024-11-14
> **Response to Reviewer Qy2H**
>
> **Thank you for your review.**
>
> Based on your comments, we politely speculate that your background might be in the traditional Computer Vision domain of AI-1.0, and you may not be a frontline researcher in the LVLM-related field. Therefore, there might be some bias in your assessment of GOT’s contributions and innovations for LVLM-OCR field. We hope that we can address your concerns to the best of our ability.
>
> **[Highlight]**
> First, we emphasize that the design of the GOT model is unique and innovative in both the fields of OCR and LVLMs. OCR performance is one of the most important capabilities in current LVLMs, and strong text perception serves as the foundational design principle for an LVLM. The introduction of GOT provides a new methodology for designing LVLMs, which is reflected in:
>
> -  A key highlight of the paper is that for LVLMs, performing dense perception tasks (such as document-level OCR) does not require as many image tokens (which is currently a bottleneck in LVLM design). GOT can decode images containing over 4000 words using only 256 image tokens, whereas other LVLMs waste thousands of tokens.
> In other words, for LVLMs, token density  (= number of encoded pixels / visual tokens, proposed in MiniCPM) still has significant room for improvement:（GOT: 4096; MiniCPM2.6: 2822; GPT4o:1088; Qwen-VL-max:784; InternVL2: 706; LLaVA-Next:157)
>
> - There may still be room to reduce the size of the decoder. We achieved a better general OCR performance than a model larger than 72B （Qwen-vl-max）using only 0.5B, providing a reference for designing small-scale LVLMs for edge devices.
>
> **[Training]**
> With regard to model training, our training architecture also has valuable and guiding significance in the LVLM field. GOT’s training primarily focuses on how to save training resources, especially when training the encoder and expanding the max token length. Our aim is completely different from decoupled pre-training approaches (such as UniT, BLIP),  because their parameters and max token length may not necessarily encounter the resource bottlenecks faced during LVLM training. The value of our training paradigm lies in:
> - To achieve a model with high token density, there should not be a freeze LLM phase in the entire LVLM training process. This is the approach taken by many existing LVLM models, such as BLIP-2, LLaVA, Qwen-VL, and mPlug-Owl in their stage-1. Instead of directly aligning image tokens to the LLM, the tokens must be aligned with each other to obtain high compression decoding ability for text.
> - The above settings will inevitably significantly increase the required resources. GOT’s stage-1 aims to use a small language model to train an encoder suitable for LLM under lower resource conditions, with the key point absolutely not being decoupling pretraining (you said UniT, BLIP). GOT’s stage-2 is designed to train data with diversity and shorter tokens, while stage-3 is for training multi-page, crop, and other extremely-length max token (e.g., 8K long texts) scenarios.
>
>  In summary, the design and data selection for each of our training stages are not aimed at performance, but rather at conserving GPU resources. Resource-saving is essential in training LVLMs, especially when dealing with extremely long texts, such as document OCR. We did not conduct ablation experiments for each stage because, given our computational resources, each stage is necessary, and the absence of any stage would prevent the completion of the training process.
>
> **[Presentation]** Due to space constraints, we did not include too many sub-sections or architectural diagrams. We are considering reorganizing the structure to enhance understanding of our model.
>
>
> **[Your question]**
> - In the LVLM field, particularly given the strong memory capabilities of LLMs, models usually choose to train for only one epoch, and our setting aligns with this prior knowledge. Multi-epoch training is only relied upon by traditional computer vision models.
> - For samples around 2000 tokens, the resource consumption for training GOT (which requires 20G memory) is much lower compared to Qwen-VL-max (which needs 1600G memory). When deployed under VLLM, GOT achieves an inference speed of approximately 1000 tokens/s on a 3090 GPU, which is several tens of times faster than the 72B Qwen-VL-max.
>
> **Sincerely hope that my response has resolved your concerns.**

---

### Note · Authors · 2024-11-24

I have read and agree with the venue's withdrawal policy on behalf of myself and my co-authors.